# Solitary and Collective Motion Behaviors of *TiO*_2_ Microrobots under the Coupling of Multiple Light Fields

**DOI:** 10.3390/mi14010089

**Published:** 2022-12-29

**Authors:** Xinjian Fan, Qihang Hu, Xin Zhang, Lining Sun, Zhan Yang

**Affiliations:** School of Mechanical and Electrical Engineering, Soochow University, No. 8, Jixue Road, Suzhou 215131, China

**Keywords:** *TiO*
_2_, photochemical microrobot, UV light driven, collective behaviors, reconfigurable microrobot swarm

## Abstract

Due to their fascinating solitary and collective behavior, photochemical microrobots have attracted extensive attention from researchers and have obtained a series of outstanding research progress in recent years. However, due to the limitation of using a single light source, the realization of reconfigurable and controllable motion behaviors of the photochemical microrobot is still facing a series of challenges. To release these restrictions, we reported a multi-light-field-coupling-based method for driving the photochemical microrobot or its swarm in a regulatable manner. Here, we first designed a control system for coupling multiple light sources to realize the programmable application of four light sources in different directions. Then a $TiO_{2}$-based photochemical microrobot was prepared, with its surface electric field distribution under different lighting conditions estimated by modeling-based simulation, where the feasibility of regulating the microrobot’s motion behavior via the proposed setup was verified. Furthermore, our experimental results show that under the action of the compound light fields, we can not only robustly control the motion behavior of a single $TiO_{2}$ microrobot but also reconfigure its collective behaviors. For example, we realized the free switching of the single $TiO_{2}$ microrobots’ movement direction, and the controllable diffusion, aggregation, the locomotion and merging of $TiO_{2}$ microrobot swarms. Our discovery would provide potential means to realize the leap-forward control and application of photochemical microrobots from individuals to swarms, as well as the creation of active materials and intelligent synthetic systems.

## 1. Introduction

Microrobots are functional devices at micro scales which can convert chemical energy, electric energy, light energy, magnetic energy, and other energy into self-driven mechanical energy. With the continuous development of robot technology, the field of microrobots has developed into a new research field. Microrobots are characterized by small size, light weight, large thrust-to-weight ratio, and flexible and controllable movement. They can adapt to different working environments and work together as a swarm. They have been widely used in different fields, such as biomedicine [1,2,3], environmental protection [4,5], micro-nano fluid control [6,7,8], and micro-nano manufacturing [9].

Moreover, as we all know, fabrication technology, motion control, swarm construction, and swarm intelligence have been the four challenges in the research field of microrobots. Among them, motion control is the key for realizing their potential applications, since it is a prerequisite for the robot to perform tasks. Nowadays, their relatively mature actuation methods and power source include chemical field [10], magnetic field [11], ultrasonic field [12], optical field [13], and biological driving mode [14]. For the optical driving technology, the external control field constructed by light is an ideal drive field, which can easily adjust the intensity of the light field by remote application of a light source. For example, near-infrared light and ultraviolet irradiation could be used to drive microrobots. Infrared light-driven micro-nanorobots can safely and effectively capture and detect circulating tumor cells (CTCS) in the entire blood environment without pretreatment. Moreover, Micro-nanorobots can cross cell membranes for drug delivery [15]; Deep penetration of tumors can be achieved [16], etc. Maric et al. studied the influence of metal layers on the motion speed of micro-nanorobots. In addition, the direction of motion of micro-nanorobots can be precisely controlled by changing the direction of ultraviolet light [17].

In terms of the motion control tendency of microrobots, researchers have gradually shifted their attention from single control to swarm control, which can satisfy more situations and tasks for their more flexible and changeable handling performance. Xie Hui’s research group used alternating magnetic fields to realize the different configurations of hematite colloidal particles in the form of chain-like, vortex-like, and ribbon-like swarms and realized the rapid and reversible conversion between them by switching different magnetic fields [18]. Sitti’s research group programmed external ferromagnet arrays to control self-repulsive ferromagnetic microrobots, created various complex static structures, and realized the cooperative operation of magnetic micron robot groups [19]. The microrobot based on sunflower pollen grains (SPG) prepared by Sun et al. will form a chain shape under the action of a rotating magnetic field, which can realize the transportation of large structures and improve their bio-inspired functionalities [20].

In addition, due to the large scanning area and uniform field effect, light field has also been widely used in the swarm control of micro/nanorobots. For instance, Mou et al. found that $TiO_{2}$ micron robots with rich hydroxyl groups could spontaneously aggregate and produce rich collective behaviors under ultraviolet irradiation, showing expansionism, negative phototaxis, and adaptive reconfigureuration [21]. Deng et al. created a micro-nano robot swarm by convective flow induced by near-infrared light and moved the swarm by moving near-infrared light spots [22]. The graphene oxide (GO) particles prepared by Qin et al. can move spontaneously to the UV light source, aggregate, and automatically and periodically contract and expand under continuous light stimulation [23]. Zhou et al. used ultraviolet light to induce hematite (${\mathrm{Fe}}_{2}O_{3}$) nanoparticles to assemble into a large complex pattern [24]. Sun et al. prepared a micron robot composed of polystyrene microspheres and polydopamine core-shell structure and controlled the aggregation shape by near-infrared light [25]. Wang et al. used blue light in pure water to drive spherical Janus microrobots based on Ag/AgCl in purified water, proving that these micron robots have specific clustering behavior and can cluster together [26]. With the continuous enhancement of motion performance, light-driven micro-nanorobots have more extensive applications, such as removing waste pollutants in water [27,28] and killing cancer cells [29].

However, the traditional driving method based on a single light source is tough to freely adjust the direction of light field gradients, which makes the robot’s motion behavior unprogrammable. Therefore such methods usually cannot actively find targets or working sites for fulfilling complex tasks. Here, we proposed a multi-light fields combined setup to obtain more controllable motion behavior of photochemical microrobots by demonstrating tactic behaviors of micromotors individually or in swarms. $TiO_{2}$ microrobot was designed and prepared by combining microrobot control technology with nanoparticle preparation technology [30]. Light can make $TiO_{2}$ microrobot move mainly based on the principle of photolysis of $H_{2}O$ [25]. The prepared micromotors under light irradiation exhibit intriguing phototaxis because of the dominant nonelectrolyte diffusiophoretic interactions, which were numerically analyzed and proved by the simulation results of the near-field intensity distribution of the resulting electric field. We then designed a series of flexible drive control technologies for the microrobot when actuated by a single or multiple UV lights, individually or collectively, based on the simulation results. Thereafter, the motion experiment of the $TiO_{2}$ microrobot and its swarm was conducted. The results show that based on the designed multi-light fields combined control system, the programmable and compound motion control of $TiO_{2}$ microrobots from individual to swarm was realized. For example, by changing the gradient of the light field, we can control the motion direction of the microrobot in situ. In addition, $TiO_{2}$ microbot swarm also exhibits richer and more attractive collective behaviors. They can integrate to form a large swarm by regrouping the scattered individuals or splitting them into decentralized individuals. Moreover, the shape and location of the swarm can also be changed through the coupling of multiple light fields, which gives them the potential to adapt to the environment according to local landscapes. Our research results will lay a foundation for the optical control system of photochemical microrobots and potentially accelerate such robots’ application process in clinical medicine, biomedicine, and other industries.

## 2. Working Principle

### 2.1. Preparation of the $TiO_{2}$ Microrobot

As shown in Figure 1a, to realize a procedure for batch preparation micron-scale photochemical robots, here, the stirring method [31] was adopted, where we added 25 mL of alcohol, 100 μL of NaCl (0.1 mol/L), and 425 μL of tetrabutyl titanate by control the feeding mode of tetrabutyl titanate in the form of drip. The magnetic agitator was used to stir it evenly for 18 min, and then the reactant was left to stand for 24 h. The prepared microrobots were collected and characterized by optical microscope and scanning electron microscope (SEM) images, respectively. Figure 1b shows the completed reaction samples collected in the reactor, namely $TiO_{2}$ particles dispersed in the experimental environment, which need to be thoroughly washed by the centrifugation method before being applied to the experiment. According to the difference between the density of $TiO_{2}$ and the density of water, a low-speed centrifuge at the speed of 1600 r/min is used. The centrifugation time is 3–4 min, and the microrobot samples can be collected at the bottom of the tube. Then the bottom samples were retained, new deionized water was added, and the above experiments were repeated several times to obtain a relatively pure $TiO_{2}$ microrobot solution. After centrifugation, the samples were placed in a glass dish at 60 ${}^{{}^{\circ}}$C to obtain the $TiO_{2}$ powder sample.

Next, the $TiO_{2}$ microrobot was characterized by an optical microscope. Figure 1c shows the optical imaging results of $TiO_{2}$ microrobots prepared by stirring method, which were sampled and placed on a slide. From evaluating these particles’ sizes in the image, it can be known that most of the microrobots were about 1 μm in size. Moreover, the specific size can be further and more accurately obtained and characterized by SEM images due to the unattainable resolution of optical microscopes. Therefore, to more clearly observe the characteristic parameters of the prepared microrobot, including its sample morphology and size, characterization based on the SEM results was conducted on the above-prepared samples. Figure 1d shows the SEM image results of $TiO_{2}$ microrobot prepared by stirring method with a nearly smooth surface, and we statistically analyzed the diameter of the particles, as shown in Figure 1e. It can be seen from the results that the size range of the robot is within 0.5–1.5 μm, mainly around 0.8 μm, since the area of the red part in this figure exceeds 65% of the total area.

### 2.2. Design of the Optical Control Platform

Figure 1f illustrates the framework of the physical composition of the optical control platform with multi-light fields. As shown, this system is mainly composed of four independent UV point light sources, an imaging unit, control software, etc. The UV point light source probe is, respectively, placed at the position with an angle of 90 degrees at the four ends to realize the adjustment of the light field gradient in the four directions of up, down, left, and right. The design parameters of the optical control platform involve the incident angle, incident power, and intensity regulation of the incident light of the point light source, etc. It should be noted that to precisely control the motion behavior of the microrobot, the incident angle of each point light source is designed to shine on the center of the working plane uniformly. Physical close-up view of the multi-light-field-based optical driving platform can be found in Figure 1g. Moreover, to ensure the control system has a suitable workspace (marked by the rectangle in Figure 1f, which is 5 cm × 5 cm in our system) for dealing with large stroke experiments, a two-axis motorized translation stage (25 cm range, 0.2 mm precision on *x*-*y* axes) was introduced to ensure that the focus of spots is always located near the region of interest.

Next, to observe the motion behavior and state of the microrobot under different lighting conditions in real-time, the experimental process was acquired and visualized with the ZEISS microscope (Axiovert 40 C). The objective lens of the optical microscope used in our experiments is 40×, and the eyepiece is 15×. Therefore, the maximum observation magnification of this bio-optical microscope is up to 600×, where $TiO_{2}$ particles with a size of about 1 μm can be observed without other external auxiliary system is required. In addition, the broad view field above the device allowed the installation of an upright microscope to obtain the position and behavior of the microrobots, with the resolution of the installed charge-coupled device (CCD) camera being 1600 × 1200 (MS23-H model, Mingmei Optoelectronics), as shown in Figure 1g. LabVIEW-based upper computer software was developed to control light intensity and light pulse time through a serial port (RS-232), enabling the microscope to establish communication with the workstation. In addition, based on the home-made software, we can also achieved the real-time acquisition and processing of camera images to record the experimental process.

### 2.3. Numerical Analysis of $TiO_{2}$ Microrobots’ Motion Behavior

The process of simulating and optimizing the design and driving strategy of the $TiO_{2}$ microrobot based on the finite element method is shown in Figure 2a. For example, to analyze the motion state and trend of $TiO_{2}$ microrobots under different lighting conditions, here, the light intensity suffered by a single $TiO_{2}$ microrobot or $TiO_{2}$ microrobot swarm was numerically estimated, and the resulting electric field intensity distribution around the swarmed $TiO_{2}$ was also analyzed under the given light irradiation. The reasons for the emergency of the $TiO_{2}$ microrobot’s collective behaviors are explored immediately. The driving mechanism of $TiO_{2}$ microrobots is based on the principle of photocatalysis, which makes them show typical phototaxis after being irradiated by UV light with a directional gradient. The detailed explanation of this phenomenon can be summarized as follows: since the structure of the prepared $TiO_{2}$ microrobot is completely symmetrical, the generated H${}_{+}$ concentration gradient changes a little, and their overall motion is controlled by the local electric field constructed by light as explained by Guo et al. [32]. Figure 2b shows the field intensity distribution on the surface of the microrobot formed by the light source irradiated along the positive direction of the x-axis. The value of the electric field intensity on the near-light side is 2.06 V/m, and that on the far-light side is 3.02 V/m. Therefore, the constructed asymmetric field enables the $TiO_{2}$ microrobot to move in the positive direction along the x-axis, which means the $TiO_{2}$ microrobot perform phototactic motion under the action of the local electric field constructed by light. Figure 2c shows the simulation data of the electric field around $TiO_{2}$ microrobot after four sides are irradiated by UV light, with all the illumination directions being along the *x*-axis and *y*-axis, respectively. By careful analysis and comparison of the simulation results, the maximum light intensity of the four-port microsphere on its face is 3.8 V/m. Therefore, by adjusting the illumination intensity in four directions, we can create specific light field conditions to realize the trajectory motion control of the microrobot in the two-dimensional plane.

The light wavelength λ of UV light is also one of the factors affecting the movement of the microrobot. Figure 2d shows the distribution of electric field intensity generated by UV light irradiation on the surface of the $TiO_{2}$ microrobot with λ= 300 nm, 360 nm, 370 nm, and 380 nm, respectively. It can be seen from the calculation results in Figure 2d that the shorter the wavelength of UV light, the more uniform the distribution of electric field intensity on the surface of the microrobot. Therefore, the change of light on the gradient field is more directional and targeted, making it easier to realize the precise trajectory control of $TiO_{2}$ microrobot. However, the shorter the wavelength is, the stronger the energy of the light source will be, which maybe introduces a redundant photothermal effect on the microrobot’s motion behaviors. Based on the analysis and conclusions drawn from the above simulation results, UV light with a wavelength of 365 nm was selected due to its excellent driving performance and negligible photothermal effect during the experiment process. Moreover, the size of the microrobot also determines the conversion efficiency of light energy. Figure 2e shows the distribution of electric field intensity values on the surface and inside of $TiO_{2}$ microrobot with a diameter of 500 nm and 200 nm under UV irradiation with light wavelength λ= 365 nm. It can be seen that the gradient of the field intensity of the microrobot with a small diameter is ignorable. For example, the result in Figure 2e shows the difference between the peak field strength and the peak-valley field strength of a 200 nm $TiO_{2}$ microsphere is only 7.26 V/m. In comparison, that of $TiO_{2}$ microspheres with diameters of 500 nm reached 28.74 V/m, indicating a much easy motion control requirement.

More interestingly, when a batch of hydroxyl-rich $TiO_{2}$ microrobots are dispersed in water, a large amount of H${}_{+}$ and OH${}_{-}$ will emerge around them, which will induce various convective electric fields (Figure 2f) to form in local space with their direction pointing inside due to the different diffusion capabilities of the H${}_{+}$ and OH${}_{-}$. As a result, many $TiO_{2}$ microrobots will be enriched on the substrate. In addition, it can be seen from Figure 2g that the electric field intensity inside the microrobot at the center of the illuminated area is the highest, with their external electric fields gradually decreasing. Taking into account our previous conclusions: the $TiO_{2}$ microrobot will move along the direction of the low external electric field when exposed to UV light, then we will conclude that the $TiO_{2}$ microrobots swarm will undergo negative phototaxis movement due to asymmetric UV light and exhibit expansion behavior during irradiation, and the swarm will be emerge again when the light source is turned off.

## 3. Results and Discussion

### 3.1. Solitary Behaviors of the $TiO_{2}$ Microrobot under Multiple Light Fields

To verify the controllability of the movement behavior of the $TiO_{2}$ microrobot monomer driven by the coupling of multiple light fields, we have carefully studied the motion law of a single $TiO_{2}$ microrobot as the schematic diagram demonstrated in Figure 3a,b, where the UV light (or no UV light) from the left and right sides of $TiO_{2}$ microrobots were alone or alternately applied. When there is no UV light, the $TiO_{2}$ robot follows Brownian motion and generates displacement in a random direction, which can be demonstrated in Figure 3c and Appendix A. However, when UV light shines from a specific direction, the robot follows a typical phototactic movement. Figure 3d shows the motion state of $TiO_{2}$ microrobot during the illumination time when the point light source irradiates on the left, and Figure 3e shows the result when the light source irradiates from the right side. The optical power density was 2600 ${\mathrm{mW}/\mathrm{cm}}^{2}$, and the irradiation time lasted about 300 s. As indicated that once the UV light is turned on, the $TiO_{2}$ microrobot will move toward the direction of the light spot, that is, the $TiO_{2}$ microrobot will move from right to left in Figure 3d, and the opposite is true in Figure 3e with the rate is about 0.5 μm/s after the left or right light source is irradiated, as demonstrated in Appendix A.

More interesting is that since we have placed four independent light sources in different directions, we can generate an illumination gradient in any direction on the working plane without moving the light source, but just through the superposition of several light sources, as illustrated in Figure 3f. Therefore, in theory, we can control the movement of microrobots in all directions based on our multi-light field composite system. Consequently, we can control the movement of microrobots in all directions based on our optical control platform with multi-light fields integrated. Figure 3f and Appendix A show the situation where we have oblique lighting gradients by turning on two adjacent light sources with the same light intensity. Obviously, the upper left corner of the picture is closer to the light source at this time, so the light is intense, and the lower right corner is weaker, thus forming a light gradient. At this time, the micro-robot will generate a negative driving light from the upper left to the lower right (45 degrees off the *x*-axis) behavior. Therefore, it can be imagined that we can further adjust the angle between the light gradient and the *x*-axis by changing the brightness relationship between the two light sources, thus realizing the directional movement of the $TiO_{2}$ microrobot monomer.

It is worth noting that light intensity is an essential regulatory factor when it comes to the motion behavior of $TiO_{2}$ microrobot. To prove that, we compared its motion behavior under different lighting conditions. According to the experimental results, the motion speed of $TiO_{2}$ microrobot decreases significantly after light intensity reduction, as proved by the comparison between Figure 3g,h. Specifically, our experimental results show that when the light intensity is 2600 ${\mathrm{mW}/\mathrm{cm}}^{2}$, the microrobot can move about 40 microns in 200 s; however, when the light intensity is reduced to 1600 ${\mathrm{mW}/\mathrm{cm}}^{2}$, the movement speed of the $TiO_{2}$ microrobot slows down significantly, and it took another 200 s to move a distance of about 15 microns, which means that its speed is directly reduced to no more than a half of the original one. This provides us with a feasible idea to qualitatively regulate the $TiO_{2}$ microrobot’s movement speed by flexibly adjusting light intensity.

### 3.2. Collective Behaviors of the $TiO_{2}$ Microrobot under Multiple Light Fields

The phenomenon of flocking movements of animals is ubiquitous in nature. These creatures tend to self-organize to form swarms, showing unique intelligent behaviors and functions that individuals do not possess, such as ant colonies, geese groups, etc. Therefore, by imitating the collective behavior of organisms, developing a swarm of microrobots is expected to enable them to complete complex tasks that a single robot cannot. Fortunately, as our above simulation results noted, $TiO_{2}$ microrobot would spontaneously form a swarm and then exhibit expansionary phototaxis motion induced by external UV light, which can be directly proved by the experimental result in Figure 4. The key to this phenomenon is that the surface of hydroxylated $TiO_{2}$ microrobots is rich in H${}_{+}$ and OH${}_{-}$. However, the diffusion rates of these two ions are inconsistent, which induces a diffusion electric field around, with the direction of the electric field directing from the external to the internal of the solution. Therefore, a converging flow field is formed near the substrate, resulting in the self-assembly of $TiO_{2}$ microrobots, as shown by the schematic diagram in Figure 4a. However, when UV light is applied with a light field gradient radiating from the center to the outside, a large amount of H${}_{+}$ will be included on the surface of these particles due to the catalytic mechanism of $TiO_{2}$, which will form an electrostatic repulsion with the original rich H${}_{+}$ on the surface and produce a fluid that expands outward. In this case, $TiO_{2}$ microrobot swarms exhibit negative phototaxis and scatter in all directions, as illustrated in Figure 4b. Therefore, the microrobot swarm will emerge with expansion-contraction style motion behavior by applying pulsed UV light. More interestingly, if the light gradient exists along a specific direction, the microrobot swarm will exhibit excellent motion controllability, where they would collectively move towards the opposite direction of the light gradient (Figure 4c). Furthermore, due to the remarkable superposition effect of the swarms, they exhibit excellent motion behaviors with movement speeds much higher than that of a single microrobot.

In order to apply this special motion mode to realize the multimodal behaviors control of the $TiO_{2}$ microrobot swarm, a series of detailed light control experiments were carried out, as shown in Figure 4d–g. Firstly, we focus multiple light sources on the center of the workspace. Before the light sources are turned on, the microrobots spontaneously assemble into swarms due to the secondary electric field. However, when the UV light sources (1600 ${\mathrm{mW}/\mathrm{cm}}^{2}$) are turned on, the individual robot inside the swarm begins to scatter in all directions. In this way, by repeating these processes, one can realize the periodic expansion-contraction style motion phenomenon, as shown in Figure 4d and Appendix A.

We then studied the trajectory control of microrobot swarms under the combined action of multiple light fields depending on our optical control platform, where the pulsed light from multiple angles is applied sequentially. When there is no UV light irradiation, the $TiO_{2}$ microrobot swarms emerge in a certain space region according to the initial distribution state of the microrobots. However, once 2600 ${\mathrm{mW}/\mathrm{cm}}^{2}$ UV light from the lower right corner was given (lights in the x+ and y− direction are lit at the same time), $TiO_{2}$ microrobot swarm began to move to the upper left corner of the region. After 100 s, UV light in the positive direction along the x− was given, and then the microrobots moved to the upper right corner of the region until the UV light in the negative direction along the y+,x+ was turned on. In this way, after repeated cycles of UV light on different sides, the $TiO_{2}$ microrobot swarm moves to the lower right corner of the area, as shown in Figure 4e and Appendix A.

In addition, it is more exciting that microrobot swarms will produce interesting fusion phenomena due to the mutual flow field during the movement. Therefore, the number of members in the swarm can be adjusted as needed, which means even a large-scale swarm can be generated to complete more complex tasks. Based on the former result, the experiments are performed in two cases. Figure 4f and Appendix A show the process of two originally independent $TiO_{2}$ microrobots approaching each other and finally blending together under the control of the compound light field. Besides benefiting from their robust collective behaviors, the $TiO_{2}$ micromotors can perform excellent parallelism and collaboration even in a large scope of the environment. As shown in Figure 4g and Appendix A, to observe more microrobot swarms at once, we adjusted the objective microscope lens to 20×. In this way, batches of $TiO_{2}$ microrobot swarms can be seen in the observed field of view. As shown in Figure 4g, here, we only apply the light field in the *y*+ direction. At this time, the formed clusters will move toward the negative direction of the *y*-axis. Due to the large number and high density of swarms in the field of view, during the process of moving in parallel, they are prone to collide with each other, showing interactive collective behavior, which gifts them with high controllability and rich motion modes, and potentially enables them to better adapt to various dynamic environments. This may pave the way for developing intelligent swarming micro/nanorobots for cooperative targeting micromanipulation and advancing their applications in drug delivery and microengineering.

## 4. Conclusions

Microrobots are functional devices that can convert chemical energy, electric energy, light energy, magnetic energy, and other energy into self-driven mechanical energy. Such kind of devices have potential applications in drug-targeted therapy, treatment of atherosclerosis, wound cleaning, auxiliary coagulation, and so on. For the optical drive technology, the external control field constructed by light is an ideal drive field that can easily adjust the intensity of the light field by remote application of a light source. Therefore, UV (ultraviolet) light is used to drive the system in this paper, and a two-dimensional optical control platform is established for the photoelectric generation effect, which improves the accuracy of trajectory control of microrobots and lays a certain research foundation for the optical control system of microrobots. The trajectory control experiment of the isotropic $TiO_{2}$ microrobot swarm was conducted according to the designed platform. According to the dynamic instruction signal sent by the upper computer, the pulse UV light on different sides was given to realize the movement behavior programmable control of $TiO_{2}$ microrobots from individual to swarm. Different from the driving system based on a single light source, the individual microrobots under UV irradiation present controllable monomer movement and rich types of collective behaviors. Moreover, in this study, only one photochemical microrobot ($TiO_{2}$) is explored. However, other photochemical microrobots will likely behave similarly as their movement behavior is only related to the spatial structure of the light field, which means this method provides a universal inspiration for controlling photochemical microrobots or their swarms. Consequently, the presented method of driving $TiO_{2}$ microrobot offers a promising approach for the robust control of photochemical micromotors, and the sequential actuation strategy by multiple light fields can further expand its utility in targeted drug delivery and lab-on-a-chip applications.

## Figures and Tables

**Figure 1 micromachines-14-00089-f001:**
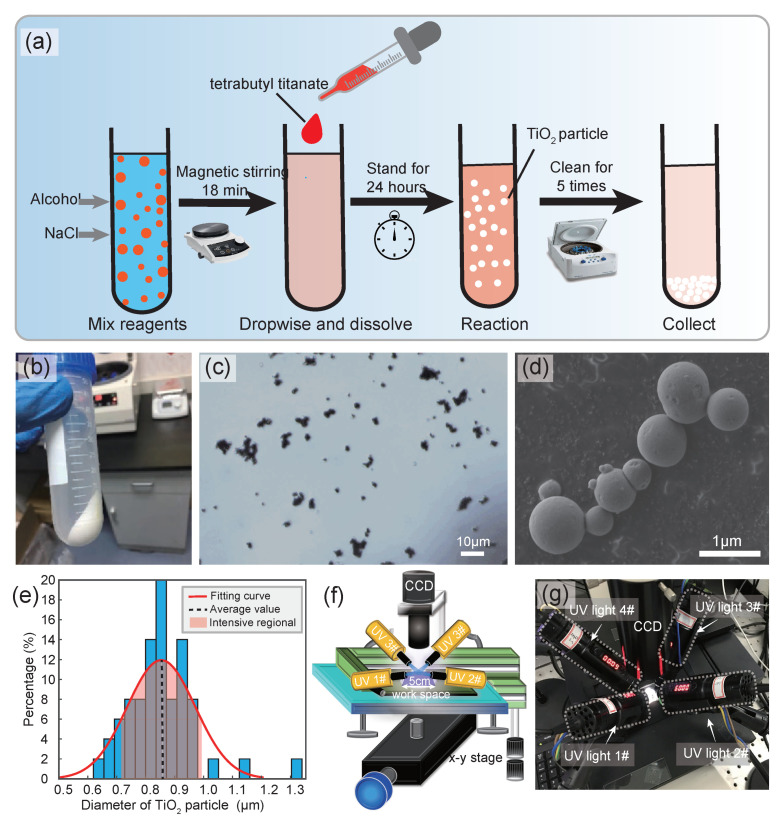
Characterization and optical control platform of the microrobots. (**a**) Schematic diagram of preparation principle and process of the $TiO_{2}$ microrobot. (**b**) Photo of the resulting $TiO_{2}$ microrobots in a centrifuge tube. (**c**) Observation results of $TiO_{2}$ microrobots under an optical microscope. (**d**) Characterization results of $TiO_{2}$ microrobot’s morphology under SEM. (**e**) Statistics and analysis results of $TiO_{2}$ microrobots’ diameters. (**f**) Schematic diagram of the multi-light-fields based optical driving platform. (**g**) A physical close-up view of the multi-light-fields-based optical driving platform.

**Figure 2 micromachines-14-00089-f002:**
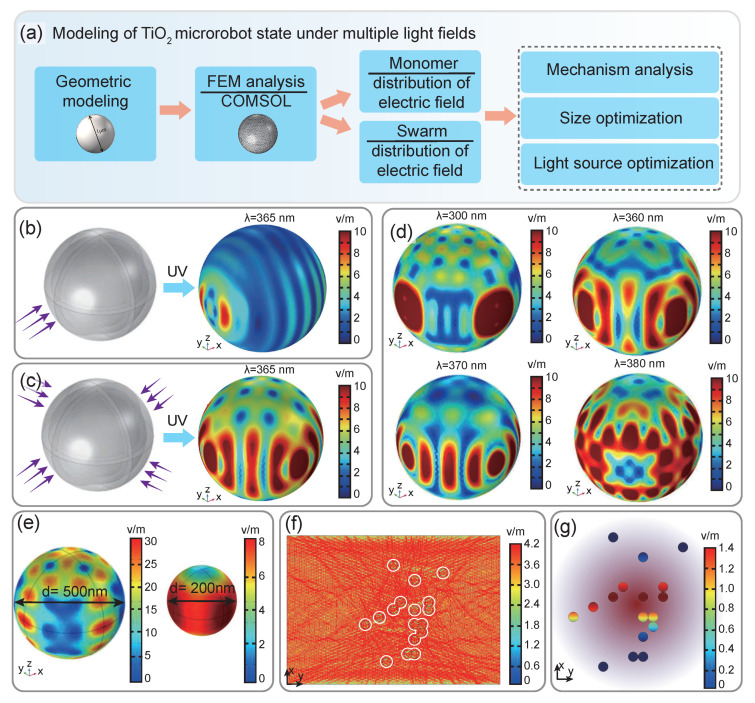
Modeling of $TiO_{2}$ microrobot motion state under multiple light fields. (**a**) The process of simulating and optimizing the design and driving strategy of the $TiO_{2}$ microrobot that based on the finite element method. Electric field distribution within a $TiO_{2}$ microrobot when irradiated by a single (**b**) and multiple (**c**) light sources. (**d**) Electric field distribution within a microrobot induced by UV light with different wavelengths λ. (**e**) Electric field distribution around $TiO_{2}$ microrobots with different diameters under the same illumination conditions. (**f**) Electric field distribution around the $TiO_{2}$ microrobot swarm when irradiated by multiple light sources. (**g**) The electric field distributions inside the individual of the $TiO_{2}$ microrobot swarm under UV light illumination.

**Figure 3 micromachines-14-00089-f003:**
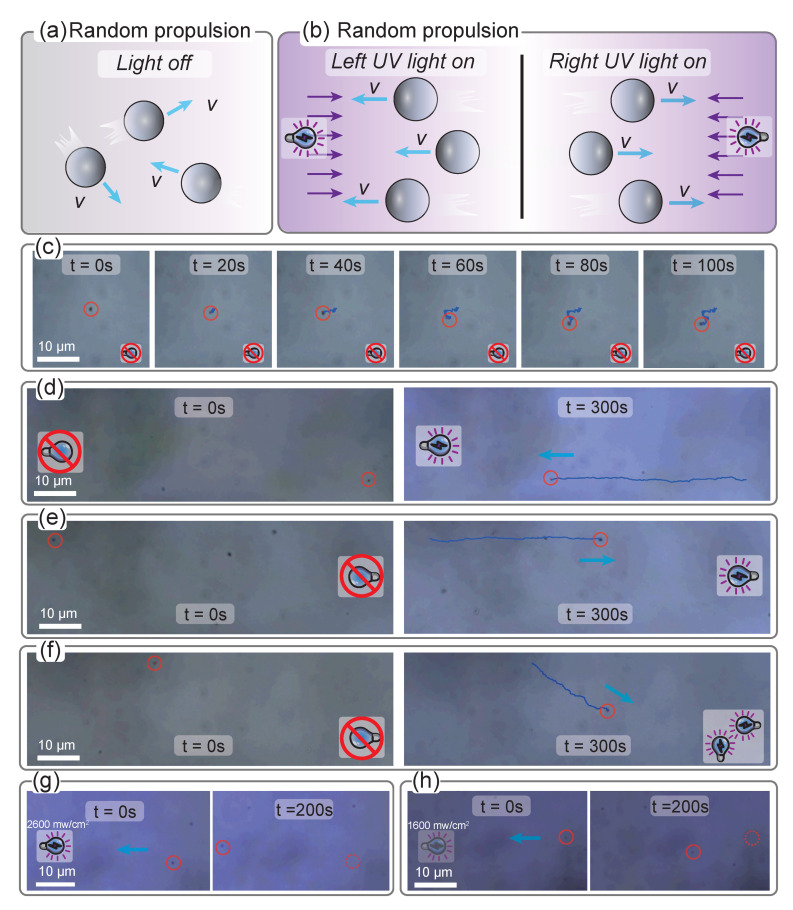
The motion behavior test experiment of a single $TiO_{2}$ microrobot under the action of multiple light fields. Schematic diagram of the motion behavior of $TiO_{2}$ microrobots with no UV light field (**a**), and UV light field from left and right (**b**), respectively. (**c**) Random propulsion experiments of the $TiO_{2}$ robot in the absence of light. The inside of the red circle is the location of the robot. Light-driven locomotion behavior of the $TiO_{2}$ microrobot under left-side (**d**) and right-side (**e**) UV illumination. Light blue arrows indicate the direction of movement. (**f**) Motion behavior of the $TiO_{2}$ microrobots under a compound UV light field with the right and lower light sources turned on. Movement of $TiO_{2}$ microrobot under a strong (**g**) or weak (**h**) UV light field with the dashed red circle indicates the initial position of the robot.

**Figure 4 micromachines-14-00089-f004:**
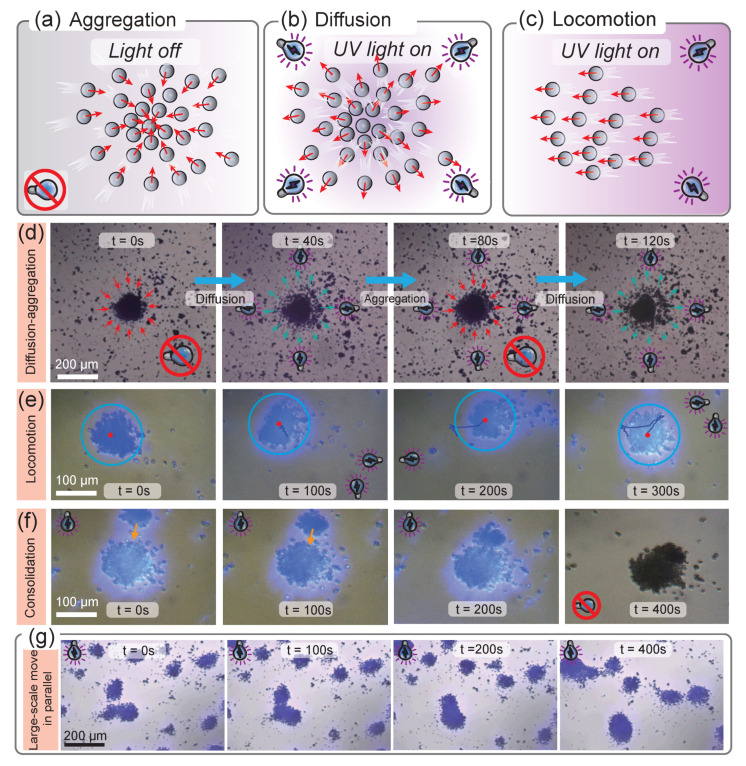
Collective behavior of $TiO_{2}$ microrobots under the action of multiple light fields. (**a**) Schematic diagram of self-aggregation of $TiO_{2}$ microrobots in the absence of light field. The red arrow indicates the direction of motion of the robot. (**b**) Scattering diagram of the $TiO_{2}$ microrobot swarm under UV light field. (**c**) Locomotion diagram of the $TiO_{2}$ microrobot swarm under directional UV light field. (**d**) Controllable aggregation and diffusion behavior of a large number of $TiO_{2}$ microrobots under pulsed UV light. The blue arrow indicates the direction of progress. (**e**) Locomotion of $TiO_{2}$ microrobot swarms under the sequential control of the multi-light source platform. The blue circle and red dot represent the position of the swarm. (**f**) Merging process of $TiO_{2}$ microrobot swarms under multiple UV illuminations. Orange arrows indicate the direction of movement of the swarm. (**g**) Combined motion and merging of large-scale microrobot swarms under the control of the multi-light source platform in parallel.

## Data Availability

All data needed to evaluate the conclusions in the paper are present in the paper and/or the Appendix A.

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
