# Peer review of "Solitary and Collective Motion Behaviors of TiO2 Microrobots under the Coupling of Multiple Light Fields"

_micromachines, 2022, doi:10.3390/mi14010089_

Round 1

Reviewer 1 Report

This manuscript presents a new strategy for autonomously manipulating a single photochemical microrobot and its swarms in liquids via a novel optical driving platform, which regulates the light field gradient inner the workspace through four independent light sources. The main contribution is that a series of light-controlled micro-robot driving strategies, from individual to swarm, were proposed and analyzed theoretically, making the photochemical microrobot movement more flexible while providing greater feasibility for good compatibility in complex environments. In addition, in order to confirm the feasibility of the system and control strategies, the authors fully demonstrate the conclusions by deploying a large number of verification experiments. In general, the ideas and developments are generally clearly described and are presented at an adequate length. Experiments are also abundant, which thoroughly verifies the highlights of the article. I particularly liked the video as a nice visual illustration of the different features and modes of locomotion. The paper is scientifically well and accurately written. The contribution seems significant. I believe it can be a good fit for the micromachines journal.

Reviewer 2 Report

In this article, the authors designed a light-controlled microrobot drive platform consisting of four UV light sources. Compared with the traditional single-drive platform, it has more controllable degrees of freedom. The drive platform can realize the spatial regulation of the gradient of the light field and then can make full use of the phototaxis of the light-controlled micro-robot to realize its trajectory control. In addition, based on this platform, the authors realized the behaviour control of light-controlled microrobot swarms and achieved good experimental results in trajectory control, scattering and merging, and large-scale parallel motion of microrobot swarms. In short, this paper has clear ideas, sufficient experimental content, and a clear theoretical analysis. It is an innovative result in the field of light-driven microrobot research. I am pleased to recommend the manuscript for publication after the authors address the following comments:

Comment 1: It seems that the workspace size of the setup is not mentioned in the manuscript. Please give the related information.

Comment 2: The authors seems to have forgotten to add the scale bar in video 3. Please add relevant information so that readers can understand easily.

Comment 3: There is a typo on line 95 of the article: “add” should be changed to “added”.
